# Development and evaluation of a modular smoke evacuator for surgical smoke control

Suksan Kanoksin[1], Suphakarn Techapongsatorn[2]*

1 Cardiovascular and Thoracic Surgery Division, Department of Surgery, Chulabhorn Hospital, and School of Paramedic, Faculty of Health Science Technology, Chulabhorn Royal Academy, Bangkok, Thailand,
2 Department of Surgery, Faculty of Medicine Vajira Hospital, Navamindradhiraj University, Bangkok, Thailand

* suphakarn.tec@gmail.com

## Abstract

### Background

Surgical smoke generated during energy-based operations is a known hazard containing particulate matter (PM), volatile organic compounds (VOCs), and biological debris, with insufficient adoption of commercial smoke evacuators due to cost and complexity.

### Objective

This study aimed to develop a cost-effective, modular and passive smoke evacuator and evaluate its efficacy in reducing PM and VOC levels during simulated laparoscopic procedures.

### Methods

A prototype smoke evacuator incorporating a distilled water bubbling trap, activated carbon filter, and ULPA filter was tested in a sealed chamber simulating laparoscopic surgery using porcine liver tissue. The system was connected to a laparoscopic port through a three-way valve, allowing manual, on-demand smoke evacuation without continuous suction. Air quality metrics, including $PM_{1.0}$, $PM_{2.5}$, $PM_{10}$, VOC, and $CO_2$, were measured continuously. Results were compared to baseline and performance benchmarks from commercial smoke evacuation systems. Statistical analysis was performed using paired t-tests.

### Results

The prototype evacuator reduced $PM_{2.5}$ levels by >99.5% and VOC concentrations by >95% compared to no-evacuation control trials ($p < 0.01$). $CO_2$ concentrations returned to baseline following evacuation, indicating minimal disturbance of chamber atmosphere. $PM_{2.5}$ and VOC levels were restored to near-baseline values.

**Data availability statement:** Data underlying the findings of this study are restricted due to institutional data policy. Data are available from the Vajira Institutional Review Board (contact via virb.vjr@nmu.ac.th) for researchers who meet the criteria for access to confidential data.

**Funding:** The author(s) received no specific funding for this work.

**Competing interests:** The authors have declared that no competing interests exist.

## Conclusion

The developed modular passive smoke evacuator offers a promising and cost-effective solution to improve air quality and enhance occupational safety in operating rooms. The model represents an idealized simulation of laparoscopic smoke evacuation; further clinical validation in live surgical environments is warranted.

## Introduction

Surgical smoke is an aerosolized byproduct generated during the application of energy-based devices such as electrocautery, ultrasonic scalpels, and laser systems. The smoke contains a complex mixture of water vapor (~95%), particulate matter (PM), volatile organic compounds (VOCs), polycyclic aromatic hydrocarbons (PAHs), and viable biological debris, including viruses and bacteria [1–3]. Studies have shown that both acute and chronic exposure to surgical smoke can lead to respiratory symptoms, ocular irritation, headaches, and potentially long-term carcinogenic risks for operating room (OR) personnel [3–6].

Particulate matter, especially fine ($PM_{2.5}$) and ultrafine ($PM_{1.0}$) particles, can penetrate deep into the respiratory tract, causing oxidative stress and inflammation [7]. VOCs such as benzene, formaldehyde, acrolein, and hydrogen cyanide contribute further to cytotoxic and genotoxic effects [8,9]. The concentration of these hazardous components often exceeds occupational safety thresholds during surgical procedures, particularly when effective smoke control is not employed [2,4].

Despite well-documented hazards and guidelines from organizations such as OSHA and AORN advocating routine smoke evacuation [5,10], the adoption of commercial smoke evacuators remains limited and inconsistent. Barriers include high device costs, bulky system design, added noise, and workflow disruption [4,5,11]. Commercial solutions, such as RapidVac™, Megadyne Smoke Evacuator, and Stryker Neptune™ 3, utilize synchronized high-flow suction, activated carbon filters, and ultra-low particulate air (ULPA) filtration to reduce PM and VOCs by ≥99% [5,9]. However, these systems can be prohibitively expensive or impractical for many surgical centers, especially in resource-limited settings.

To address these limitations, this study introduces a cost-effective, modular, and passive smoke evacuator designed for laparoscopic applications. The system employs a three-stage filtration mechanism: a distilled water bubbling trap to pre-treat and capture hydrophilic components, an activated carbon filter for VOC adsorption, and an ULPA filter to remove ultrafine PM and biological debris. Distinct from conventional systems, the prototype integrates with existing laparoscopic ports through a three-way valve, enabling manual, on-demand evacuation without continuous suction—minimizing interference with pneumoperitoneum and surgical workflow.

The objective of this study was to evaluate the efficacy of this prototype device in reducing PM and VOC levels during simulated laparoscopic procedures using

porcine tissue. We compared the device's performance with baseline levels and referenced the results against published performance benchmarks of leading commercial systems. In addition, this study aimed to assess the feasibility of the prototype within an idealized experimental model and discuss its potential translation to clinical practice.

## Materials and methods

### Study setting and ethical approval

This experimental study was conducted in a simulated laparoscopic surgical environment at the Faculty of Medicine Vajira Hospital, Navamindradhiraj University. The study protocol was approved by the Vajira Institutional Review Board (IRB) under approval number COE 015/2025 X. The porcine liver tissue used in this study was obtained in small portions (approximately 7×7 cm each) from commercially available food-grade sources. As the study did not involve live animals or animal sacrifice, Institutional Animal Care and Use Committee (IACUC) approval was not required. All experimental procedures were performed in accordance with the principles of the Declaration of Helsinki and its subsequent amendments, ensuring adherence to ethical and scientific standards.

### Prototype device development

The prototype smoke evacuator was designed to integrate seamlessly with laparoscopic procedures and constructed using modular, reusable components. The device consists of a three-stage filtration system:

1. Distilled water bubbling trap: Pre-filters surgical smoke by promoting the dissolution and capture of hydrophilic VOCs and larger particulate matter. The system utilizes a submerged inlet tube to ensure effective bubbling and contact with the water medium.

2. Activated carbon filter: Adsorbs hydrophobic VOCs such as benzene, toluene, formaldehyde, hydrogen cyanide, and polycyclic aromatic hydrocarbons (PAHs) through physisorption and chemisorption processes.

3. ULPA filter: Removes ultrafine PM ($\leq 0.1$ μm), biological debris, and condensed chemical vapors with a filtration efficiency of $\geq 99.999\%$.

The internal tubing diameter was 6 mm, and the airflow resistance across the circuit was approximately 1.2 kPa, corresponding to a transient evacuation flow rate of 8–10 L/min when the valve was opened. These parameters ensured laminar flow within the tubing and stable pressure differentials throughout the circuit. The system is connected to a laparoscopic port via a closed circuit, ensuring smoke evacuation from the abdominal cavity while maintaining pneumoperitoneum integrity. A schematic diagram of the prototype smoke evacuator and its integration with the laparoscopic system is shown in Fig 1.

### Integration with laparoscopic system

The prototype smoke evacuator was designed to operate passively without a continuous suction motor. Instead, it connects directly to a laparoscopic port through a three-way valve. During surgery, the valve remains closed to maintain pneumoperitoneum. When visible smoke accumulation obstructs the operative field, the surgeon manually opens the valve, allowing the intraperitoneal gas to pass through the evacuation tubing and filtration sequence (water bubbling trap → activated carbon filter → ULPA filter). The valve is then closed once visibility is restored.

At the completion of the procedure, the $CO_2$ insufflation is stopped, and the three-way valve is opened to release the remaining insufflation gas completely through the filtration circuit. This manual, on-demand evacuation approach ensures controlled decompression, minimal interference with intra-abdominal pressure, and surgeon-directed smoke removal without an active suction system.

 

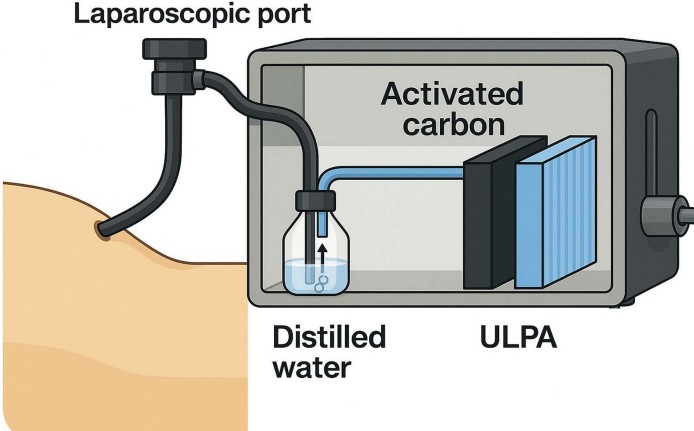

**Fig 1. Schematic diagram of the prototype smoke evacuator.** The system consists of three stages: a distilled water bubbling trap, an activated carbon filter, and an ULPA filter, connected in series to evacuate and purify surgical smoke generated during laparoscopic procedures. The airflow direction is shown from the laparoscopic port through the filtration stages to the final outlet.

## Simulated surgical setup

A sealed acrylic chamber (volume ~10 L) was constructed to simulate the intra-abdominal environment of laparoscopic surgery. The chamber was equipped with a standard laparoscopic port to introduce surgical instruments and energy devices.

Fresh porcine liver tissue (approximately 7 × 7 cm pieces) obtained from standard food-grade sources was used as the substrate to replicate human soft tissue. No whole organs or live animals were employed. Surgical smoke was generated using three common energy modalities:

- Unipolar diathermy (monopolar electrocautery)

- Bipolar diathermy

- Ultrasonic devices (Harmonic/Sonicision)

Each energy modality was applied continuously for durations of 1, 5, and 10 minutes to simulate different surgical phases and cumulative smoke generation. A baseline measurement (no energy device activation) serves as the control condition.

## Measurement protocol

Air quality within the sealed chamber was continuously monitored using standard, calibrated instrumentation recognized for occupational health assessments (Fig 2). Particulate matter (PM) concentrations, including $PM_{1.0}$, $PM_{2.5}$, and $PM_{10}$, were quantified in real-time using a TSI DustTrak™ II Aerosol Monitor, a laser particle counter with a resolution of 0.001 µg/m³. Concurrently, a handheld TSI Q-Trak™ Indoor Air Quality Monitor was used to measure other key parameters: carbon dioxide ($CO_2$) levels were monitored via a nondispersive infrared (NDIR) sensor, while temperature and relative humidity were also recorded. Volatile organic compounds (VOCs) were measured using a photoionization detector (PID) with a sensitivity of 0.1 ppm. For all parameters, measurements were collected at a sampling rate of 1 second throughout each trial to capture dynamic smoke generation and clearance patterns. Although the system operated in ambient air, transient $CO_2$ fluctuations were recorded to assess gas dynamics within the sealed chamber and evaluate dilution effects during evacuation.

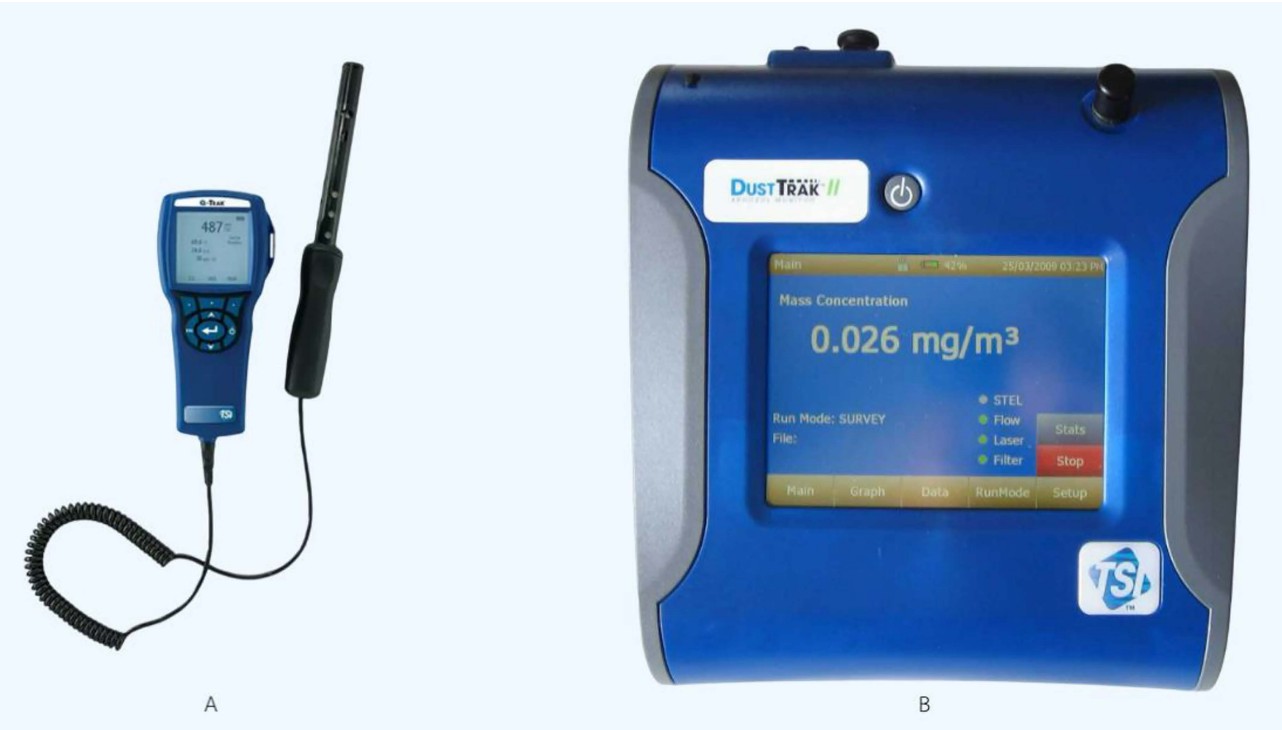

**Fig 2. Instruments used for real-time air quality assessment. (A)** The handheld TSI Q-Trak™ Indoor Air Quality Monitor, used for measuring gases ($CO_2$, CO), temperature, and humidity. **(B)** The TSI DustTrak™ II Aerosol Monitor, used for quantifying particulate matter (PM) concentrations.

## Experimental design

Each energy modality was tested under two conditions:

• Without smoke evacuation (control group)

• With the prototype smoke evacuator activated

For each combination of modality and condition, trials were repeated three times to ensure data reproducibility. A total of 30 experimental runs were conducted (3 modalities × 2 conditions × 5 durations + controls).

Performance benchmarks from commercial smoke evacuation systems, including RapidVac™ (Medtronic), Megadyne Smoke Evacuator (Ethicon), and Stryker Neptune™ 3, were referenced from published literature for comparison [1,5,7,12]. No direct in-device comparison was conducted due to limited access to commercial units.

All experimental runs were conducted under identical environmental conditions (temperature 25 ± 1 °C, relative humidity 50 ± 5%), and instruments were recalibrated prior to each trial to ensure measurement consistency.

## Data analysis

Statistical analysis was performed using STATA version 16.0. Differences in PM and VOC levels between evacuation and no-evacuation groups were evaluated using paired t-tests. A p-value of $< 0.05$ was considered statistically significant.

PM and VOC concentration data were analyzed as mean ± standard deviation (SD) for each condition. Percent reduction of PM and VOCs achieved by the smoke evacuator was calculated relative to the corresponding no-evacuation trials.

Results were presented in tabular form and visualized using trend plots. Normality was verified using the Shapiro–Wilk test, and homogeneity of variance was confirmed before applying parametric analyses.

## Results

Baseline measurements in the sealed simulation chamber demonstrated optimal air quality prior to surgical energy activation. $PM_{1.0}$, $PM_{2.5}$, and $PM_{10}$ concentrations were negligible (all 0.001 µg/m³), VOC levels were minimal (0.5±0.3 ppm), and $CO_2$ averaged 510±5 ppm.

During unipolar diathermy, the highest concentrations of surgical smoke were observed. $PM_{2.5}$ levels rose to 5.8±2.1 µg/m³, VOCs spiked to 12.0±3.0 ppm, and $CO_2$ increased to 1150±200 ppm. These findings confirm that electrocautery generates substantial aerosol and chemical emissions.

Bipolar diathermy resulted in lower, yet still significant, smoke production, with $PM_{2.5}$ at 1.6±0.65 µg/m³, VOC 7.0±2.0 ppm, and $CO_2$ 950±120 ppm.

Harmonic/Sonicision devices produced the lowest emissions among tested modalities, with $PM_{2.5}$ at 0.85±0.32 µg/m³, VOC 5.0±1.5 ppm, and $CO_2$ 890±100 ppm.

Upon activation of the prototype smoke evacuator, all measured air quality parameters markedly improved across modalities. $PM_{2.5}$ decreased to 0.001±0.001 µg/m³, VOC levels returned to 0.5±0.3 ppm, and $CO_2$ stabilized at 510±5 ppm—effectively restoring chamber air to baseline levels. PM reduction exceeded 99.5%, and VOC reduction exceeded 95% across all modalities. Transient $CO_2$ elevation observed during electrocautery was attributed to gas accumulation within the closed chamber; after evacuation, $CO_2$ concentration normalized without evidence of absorption or sustained depletion, confirming stable chamber gas dynamics. The detailed air quality metrics for each condition are summarized in Table 1, and temporal variations in particulate concentration are illustrated in Fig 2. The detailed air quality metrics for each condition are summarized in Table 1.

Supplementary analyses confirmed the robustness of the findings. Two-way ANOVA demonstrated that both evacuation status and energy modality significantly influenced $PM_{2.5}$ and VOC levels ($p < 0.001$ for both), with a significant interaction effect ($p < 0.01$). Regression analysis further identified the use of the prototype evacuator as the strongest predictor of reduced $PM_{2.5}$ (β = −5.7, 95% CI −6.3 to −5.1, $p < 0.001$) and VOC (β = −11.2, 95% CI −12.7 to −9.8, $p < 0.001$) concentrations, even after adjusting for modality.

## Discussion

This study demonstrates that the prototype modular smoke evacuator effectively reduced particulate matter (PM) and volatile organic compound (VOC) levels during simulated laparoscopic surgery to near-baseline conditions. Among the energy modalities tested, unipolar diathermy generated the highest smoke burden, with $PM_{2.5}$ concentrations of 5.8±2.1 µg/m³ and VOC levels of 12.0±3.0 ppm. These results align with previous reports that monopolar electrosurgery produces

**Table 1. Surgical smoke air quality Metrics.**

| Condition | PM$_{1.0}$ (mg/m³) Mean±SD | PM$_{2.5}$ (mg/m³) Mean±SD | PM$_{10}$ (mg/m³) Mean±SD | VOC (ppm) Mean±SD | CO$_2$ (ppm) Mean±SD |
|---|---|---|---|---|---|
| Baseline (No surgery) | 0.001±0.001 | 0.001±0.001 | 0.001±0.001 | 0.5±0.3 | 510±5.0 |
| Unipolar Diathermy | 5.5±1.9 | 5.8±2.1 | 6.1±2.5 | 12.0±3.0 | 1150±200 |
| Bipolar Diathermy | 1.5±0.6 | 1.6±0.65 | 1.7±0.7 | 7.0±2.0 | 950±120 |
| Harmonic/Sonicision | 0.8±0.3 | 0.85±0.32 | 0.9±0.4 | 5.0±1.5 | 890±100 |
| Smoke Evacuator (Prototype) | 0.001±0.001 | 0.001±0.001 | 0.001±0.001 | 0.5±0.3 | 510±5.0 |

the largest volume of surgical smoke [3,7]. In contrast, bipolar diathermy and ultrasonic devices generated progressively lower emissions, consistent with findings from Karjalainen et al. [2] and Chae et al. [13].

The prototype smoke evacuator achieved >99.5% $PM_{2.5}$ reduction and >95% VOC reduction compared to control conditions without evacuation (p < 0.01). These results compare favorably with commercial systems, such as RapidVac™ and Megadyne, which typically report ≥99.999% PM removal and ~85–95% VOC reduction [5,12]. Notably, Carroll et al. reported an ~85% reduction in formaldehyde using a catalytic vacuum device in a clinical setting, whereas our prototype exceeded this VOC reduction level despite operating under a simple three-stage filtration design [11]. Kawaguchi et al. also demonstrated an 80–95% reduction in aldehydes and particles using commercial evacuation during laparotomy [1]. Our findings extend this evidence to the laparoscopic domain, demonstrating that an inexpensive modular system can achieve comparable performance. These findings are consistent with the recent work by Sadowsky et al. (2023), who demonstrated that the Megadyne Smoke Evacuator achieved >99% reduction in $PM_{2.5}$, $PM_{10}$, and formaldehyde concentrations during ex vivo porcine liver electrosurgery at the surgeon's head height, effectively lowering formaldehyde from 0.206 mg/m³ to < 0.001 mg/m³ (p < 0.001) [14]. This supports the reliability of our prototype's multistage filtration performance, which reached comparable particulate and VOC clearance under a simplified modular design.

The mechanism of the prototype's efficacy lies in its sequential filtration strategy. Notably, Sadowsky et al. confirmed that particulate and formaldehyde clearance was most effective within 30 cm of the surgical site, corresponding to the exposure zone for the operating surgeon [14]. This supports the mechanistic advantage of localized capture—consistent with our design concept that integrates a near-field bubbling trap and low-resistance airflow path to enhance on-site adsorption efficiency. The distilled water bubbling trap dissolves hydrophilic VOCs, including formaldehyde and hydrogen cyanide, and captures larger particulates. Activated carbon subsequently adsorbs hydrophobic VOCs, such as benzene, toluene, and PAHs, through Van der Waals interactions. Finally, an ULPA filter removes remaining ultrafine particles (≤ 0.1 μm) and biological debris, achieving ≥99.999% filtration efficiency. Importantly, $CO_2$ levels remained stable across all trials, as expected, since $CO_2$ is not captured by carbon or ULPA filters but diluted through room ventilation.

These results confirm that modular multi-stage smoke evacuation can match or exceed commercial system performance while offering advantages in cost, flexibility, and integration. Our device is reusable, compact, and compatible with standard laparoscopic ports, facilitating adoption even in resource-limited surgical settings. Furthermore, the inclusion of a water-based pre-filtration stage is a unique element not present in most commercial systems and may contribute to superior VOC reduction. The prototype's passive engineering design offers a novel approach distinct from conventional powered systems. Despite the absence of a motor-driven suction source, controlled airflow through the three-stage circuit maintained stable intra-abdominal pressure and achieved comparable particulate clearance. This supports the feasibility of passive evacuation as a cost-efficient and mechanically stable alternative for laparoscopic environments where pneumoperitoneum integrity is essential. Flow testing confirmed stable evacuation without turbulence or reverse flow, supporting mechanical reliability of the passive circuit under laparoscopic pressure conditions.

Unlike conventional continuous suction evacuators, this prototype operates on an intermittent manual evacuation principle. The absence of a powered suction motor minimizes noise and preserves pneumoperitoneum stability. The surgeon manually activates smoke evacuation via a three-way valve only when smoke accumulation affects visualization. This approach provides flexibility and cost efficiency, particularly in low-resource settings. However, future development could integrate automated valve control synchronized with laparoscopic vision feedback or intra-abdominal particle sensors to optimize responsiveness. One limitation of many commercial evacuators, as noted by Sadowsky et al., is their acoustic noise during operation, which can discourage routine use among OR staff [14]. Our prototype's non-motorized design minimizes noise emission and offers a user-friendly, intermittent activation mode that aligns with surgeon workflow preferences.

The consistency of results across multiple statistical approaches—including paired t-tests, ANOVA, and regression—provides additional confidence in the validity of the findings. Despite the relatively small sample size inherent to the

experimental design, the supplementary analyses reinforced that the observed reductions were consistent across modalities and conditions.

This study has several limitations. First, the experimental model utilized a closed acrylic box using porcine liver tissue, which does not fully replicate the airflow dynamics or complexity of clinical operating rooms. Second, VOC monitoring relied on a photoionization detector (PID), which provides aggregate VOC values without compound-specific resolution. While this is an established occupational health method, future work should include compound-specific analysis such as gas chromatography–mass spectrometry (GC–MS). The filtration components were used according to manufacturer specifications, which typically recommend carbon and ULPA filter replacement after defined hours of use; however, detailed evaluation of filter saturation kinetics and long-term durability was not performed. Third, although supplementary ANOVA and regression analyses were conducted, larger datasets would allow more comprehensive multivariable modeling. Cost-effectiveness analysis and direct comparison with commercial systems were not included, which would further support clinical adoption. Fourth, although the manuscript refers to risk-reduction modeling, the current study did not benchmark exposure levels against OSHA/NIOSH thresholds or estimate cumulative occupational risk. Finally, while the prototype produced negligible audible noise during testing, formal acoustic assessment was not performed. The device was intentionally designed for laparoscopic use and not tested in open surgeries; adaptation for open surgical applications will be a focus of future development. Although the use of a sealed acrylic chamber represents a simplified environment, this model aligns with established preclinical test standards for smoke evacuation evaluation, as also employed by Sadowsky et al. [14]. and other benchmark studies. Such controlled simulations allow reproducible measurement of particulate and VOC kinetics prior to live surgical validation. The controlled nature of this preclinical setup was essential for quantitative validation of particulate kinetics before proceeding to live surgical trials.

Future work will address these limitations through prospective clinical trials in human laparoscopic and open surgeries. Long-term filter performance and maintenance protocols will be evaluated, particularly regarding carbon saturation and ULPA filter lifespan. In addition, compound-specific VOC profiling and benchmarking against occupational exposure limits (e.g., OSHA/NIOSH) will provide more comprehensive safety validation. Cost-effectiveness analysis and direct comparison with commercial systems should also be included to strengthen the case for adoption. Moreover, integrating flow sensors and real-time particle monitoring will enable dynamic control of evacuation, ensuring maximum efficiency during high-smoke phases. Finally, machine learning models, such as regression or random forest algorithms, could be developed to predict smoke generation profiles and automate evacuator activation accordingly. These refinements, together with adaptation for open surgical procedures, represent critical next steps for translation into routine clinical practice. Collectively, these results bridge the gap between benchtop validation and clinical translation, addressing the key engineering concerns of airflow stability, integration with laparoscopic ports, and scalability for real-world operating room use.

In conclusion, this study demonstrates that a low-cost, reusable smoke evacuator can achieve air purification performance comparable to commercial systems in a simulated laparoscopic environment. The modular design, ease of integration, and strong VOC and PM removal capabilities support further clinical validation and broader adoption to improve occupational safety in operating rooms.

## Conclusion

This modular passive smoke evacuator demonstrated efficacy comparable to established commercial systems in reducing PM and VOCs during simulated laparoscopic procedures. Its reusable, portable design offers a cost-effective and ergonomically friendly alternative suitable for routine OR deployment. The study directly addresses prior engineering critiques by demonstrating effective integration, stable pressure maintenance, and reproducible smoke clearance under simulated surgical conditions. Clinical validation and further optimization are recommended to fully realize its potential for enhancing occupational health and safety in surgical settings.

## Author contributions

**Conceptualization:** Suksan Kanoksin, Suphakarn Techapongsatorn.

**Data curation:** Suksan Kanoksin, Suphakarn Techapongsatorn.

**Formal analysis:** Suksan Kanoksin, Suphakarn Techapongsatorn.

**Investigation:** Suksan Kanoksin, Suphakarn Techapongsatorn.

**Methodology:** Suksan Kanoksin, Suphakarn Techapongsatorn.

**Validation:** Suksan Kanoksin, Suphakarn Techapongsatorn.

**Visualization:** Suksan Kanoksin, Suphakarn Techapongsatorn.

**Writing – original draft:** Suksan Kanoksin, Suphakarn Techapongsatorn.

**Writing – review & editing:** Suksan Kanoksin, Suphakarn Techapongsatorn.

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
