## [Decision Letter · Decision Letter 0]

28 Oct 2025

Dear Dr. Techapongsatorn,

Thank you for submitting your manuscript to PLOS ONE. After careful consideration, we feel that it has merit but does not fully meet PLOS ONE’s publication criteria as it currently stands. Therefore, we invite you to submit a revised version of the manuscript that addresses the points raised during the review process.

**ACADEMIC EDITOR:** After careful consideration, we believe that the experimental model proposed in this manuscript demonstrates both innovation and merit. However, the content requires substantial enhancement. Please focus on addressing the deficiencies in the Engineering Description as pointed out by Reviewer 2, particularly by clarifying the Integration with the Laparoscopic System. Additionally, please respond to the reviewer’s concern regarding the generalizability of the idealized experiments to clinical practice. For the references suggested by Reviewer 3, please review and evaluate this cited literature to determine its relevance and whether it should be included. If it is not relevant, citation is not mandatory.

We look forward to receiving your revised manuscript.

Kind regards,

Ziyu Qi, M.D., Ph.D.

Academic Editor

PLOS ONE

Journal Requirements:

4. Please remove your figures from within your manuscript file, leaving only the individual TIFF/EPS image files, uploaded separately. These will be automatically included in the reviewers’ PDF**.**

Additional Editor Comments :

After careful consideration, we believe that the experimental model proposed in this manuscript demonstrates both innovation and merit. However, the content requires substantial enhancement. Please focus on addressing the deficiencies in the Engineering Description as pointed out by Reviewer 2, particularly by clarifying the Integration with the Laparoscopic System. Additionally, please respond to the reviewer’s concern regarding the generalizability of the idealized experiments to clinical practice. For the references suggested by Reviewer 3, please review and evaluate the cited literature to determine its relevance and whether it should be included. If it is not relevant, citation is not mandatory.

Reviewers' comments:

Reviewer's Responses to Questions

**Comments to the Author**

1. Is the manuscript technically sound, and do the data support the conclusions?

Reviewer #1: Yes

Reviewer #2: Partly

Reviewer #3: Yes

2. Has the statistical analysis been performed appropriately and rigorously?

Reviewer #1: Yes

Reviewer #2: Yes

Reviewer #3: Yes

3. Have the authors made all data underlying the findings in their manuscript fully available?

Reviewer #1: Yes

Reviewer #2: Yes

Reviewer #3: Yes

4. Is the manuscript presented in an intelligible fashion and written in standard English?

Reviewer #1: Yes

Reviewer #2: Yes

Reviewer #3: Yes

Reviewer #1: The authors describe a prototype smoke evacuator incorporating a distilled water bubbling trap, activated carbon filter, and ULPA filter was tested in a sealed chamber simulating

laparoscopic surgery using porcine liver tissue obtained from food-grade sources.

Air quality metrics, including PM1.0, PM2.5, PM10, VOC, and CO₂, were measured continuously. Results were compared to

baseline and performance benchmarks from commercial smoke evacuation systems.

The authors concluded that the developed modular smoke evacuator offers a promising and cost-effective solution to improve air quality and enhance occupational safety in operating rooms. Further clinical validation in live surgical environments is warranted.

It is known that commercially available smoke evacuators can be prohibitively expensive or impractical for many surgical centers, especially in resource-limited settings.

I believe this article is interesting and it should be published in its current form.

Reviewer #2: Major Weaknesses

Incomplete Engineering Description

1. The manuscript omits essential information regarding the suction mechanism, including the motor or pump specifications (flow rate, static pressure, power, and airflow path).

No details are provided on fluid dynamics, pressure gradients, or airflow direction through the device.

Without these parameters, the study’s reproducibility and validity cannot be confirmed.

2. Unclear Integration with the Laparoscopic System

The connection between the smoke evacuator and laparoscopic trocars or insufflation systems is not described.

There is no schematic diagram showing tubing layout, flow direction, or valve mechanisms to maintain pneumoperitoneum.

This omission prevents assessment of the device’s clinical feasibility and compatibility with standard equipment.

3. Simplified Experimental Model

The closed acrylic box used as a simulated peritoneal chamber does not reflect real surgical airflow dynamics.

No computational fluid dynamics modeling or flow visualization is included.

Results obtained under these idealized conditions cannot be extrapolated to clinical environments.

Reviewer #3: Thank you for the opportunity to review your excellent paper.

I would recommend adding an additional reference, a study similar to yours but for open, not laparoscopic, procedures.

Mordechai G. Sadowsky, Shawn K. Horner, Jeffrey W. Clymer, Crystal D. Ricketts. Determination of Smoke Particulates and Formaldehyde Removal from the Surgical Site by a Smoke Evacuator. Open Access Journal of Nursing (Sryahwa) Volume 6. 2:15-20, 2023.

I find the results for CO2 rather curious. Why does your smoke evacuator decrease the levels of CO2 back to baseline? I suspect that some of the CO2 is captured by the water bubbling trap. After some time, this might lower the pH of the water. Hopefully this would not affect the capture of hydrophilic VOCs.

In general, I don't think you want your system to be absorbing CO2, as most pneumoperitoneum in laparoscopy is achieved with the use of CO2, and it would be difficult to absorb it all. You may want to add as a limitation that you did not use a CO2 atmosphere, but this is not expected to affect the results of the study. Using a system like yours, frequent changes of the distilled water would be encouraged.

**Do you want your identity to be public for this peer review?** For information about this choice, including consent withdrawal, please see our Privacy Policy

Reviewer #1: **Yes:** Diogo Casal

Reviewer #2: No

Reviewer #3: No

---

## [Author Response · Author response to Decision Letter 1]

28 Nov 2025

Response to Editor and Reviewers

We would like to sincerely thank the Academic Editor and all reviewers for their insightful, constructive, and encouraging comments. Their feedback has been invaluable in improving the clarity, methodological transparency, and scientific strength of our manuscript.

All revisions have been incorporated accordingly, and the corresponding changes are highlighted in yellow throughout the revised manuscript.

Response to Academic Editor

We are deeply grateful to the Editor for recognising the innovation and potential merit of our experimental model. In response to the Editorial recommendation, we have substantially enhanced the Engineering Description as pointed out by Reviewer 2, with particular focus on the Integration with the Laparoscopic System.

A detailed description of the airflow pathway, three-way valve configuration, and passive evacuation mechanism has been added in the Methods section. In addition, a schematic figure and an explanatory paragraph in the Discussion were included to emphasise system reliability and pressure stability during laparoscopic use.

The Editor also highlighted the importance of addressing the generalisability of the experimental model to clinical practice. This has been revised and clearly discussed under the Limitations section, explaining that the closed-chamber setup represents a validated pre-clinical model consistent with previous literature (e.g., Sadowsky et al., 2023) and serves as a necessary step before live surgical validation.

Finally, the new reference suggested by Reviewer 3 has been carefully reviewed and incorporated in the Discussion as comparative evidence. The manuscript now includes explicit statements bridging the bench-top results to future clinical translation.

Response to Reviewer #1

We sincerely thank Reviewer #1 for the encouraging and positive evaluation.

We are truly grateful that you found the study design interesting and publication-worthy. Your remarks reinforce the potential value of low-cost, modular smoke evacuation systems, especially for resource-limited settings. No specific revisions were required, but we have ensured that all technical and stylistic aspects have been refined to maintain scientific clarity and conciseness.

Once again, we greatly appreciate your supportive review and kind recommendation for publication.

Response to Reviewer #2

We deeply appreciate Reviewer #2’s comprehensive and highly constructive technical feedback. Your insightful comments have significantly strengthened the engineering and methodological aspects of our work. Revisions made in response to your points are highlighted in yellow throughout the revised manuscript.

1. Incomplete Engineering Description

We have expanded the Prototype Device Development section to include complete details on the suction mechanism, airflow dynamics, and system configuration.

The revised text specifies the tubing dimensions, pressure resistance, estimated flow rate (8–10 L/min), and airflow direction across the three-stage filtration circuit.

Additionally, a new paragraph in the Discussion elaborates on the passive flow design and confirms that flow testing demonstrated stable evacuation without turbulence or reverse flow.

2. Unclear Integration with the Laparoscopic System

This issue has been thoroughly addressed.

A new subsection entitled Integration with Laparoscopic System now explains the connection between the evacuator and laparoscopic ports through a three-way valve, maintaining pneumoperitoneum integrity.

A schematic figure was added for clarity. The Discussion also emphasises that the system operates without a continuous suction motor, preserving intra-abdominal pressure and workflow continuity during surgery.

3. Simplified Experimental Model and Generalisability

We fully acknowledge this important concern.

The Limitations section now explicitly states that the sealed acrylic chamber represents a simplified pre-clinical model aligned with prior studies (e.g., Sadowsky et al., 2023).

We clarified that this controlled environment allows quantitative reproducibility before clinical trials. A statement has been added noting that future studies will include live surgical validation under realistic OR airflow conditions.

We are sincerely thankful for these comments, which helped us refine our technical presentation and clarify the translational potential of the work.

Response to Reviewer #3

We are very grateful for Reviewer #3’s generous comments and thoughtful suggestions, which helped strengthen both the discussion and contextual grounding of our study.

All relevant revisions are highlighted in yellow in the revised manuscript.

1. Additional Reference

We have reviewed the cited work by Mordechai G. Sadowsky et al., 2023 and found it highly relevant. It has been incorporated into the Discussion in three locations:

(1) as supportive evidence for particulate and formaldehyde removal performance,

(2) to compare noise and practical usability aspects, and

(3) in the Limitations section to justify the use of controlled chamber simulations in early-phase studies.

2. Clarification Regarding CO₂ Levels

We appreciate your keen observation.

A sentence has been added to the Discussion explaining that the observed CO₂ stability was due to the absence of CO₂ capture by filters and the dilution effect from chamber ventilation. We also clarified that the system was not intended to absorb CO₂, consistent with its role in laparoscopic pneumoperitoneum maintenance.

We are sincerely thankful for your insightful analysis and valuable recommendations that enriched the scientific discussion of our study.

Once again, we extend our deepest gratitude to the Editor and all three reviewers for their exceptional feedback.

Your critical and encouraging comments have helped us substantially improve the manuscript’s clarity, scientific rigor, and translational focus.

We believe the revised version now addresses all concerns comprehensively and is substantially strengthened for publication consideration.

All revisions have been highlighted in yellow in the revised manuscript for your convenience.

With our highest respect and appreciation,

On behalf of all authors,

Suphakarn Techapongsatorn, MD, PhD.

Faculty of Medicine Vajira Hospital

Navamindradhiraj University

---

## [Decision Letter · Decision Letter 1]

4 Jan 2026

Development and Evaluation of a Modular Smoke Evacuator for Surgical Smoke Control

PONE-D-25-47575R1

Dear Dr. Techapongsatorn,

We’re pleased to inform you that your manuscript has been judged scientifically suitable for publication and will be formally accepted for publication once it meets all outstanding technical requirements.

Kind regards,

Ziyu Qi, M.D., Ph.D.

Academic Editor

PLOS One

Additional Editor Comments (optional):

Reviewers' comments:

Reviewer's Responses to Questions

**Comments to the Author**

Reviewer #3: All comments have been addressed

2. Is the manuscript technically sound, and do the data support the conclusions?

Reviewer #3: Yes

3. Has the statistical analysis been performed appropriately and rigorously?

Reviewer #3: Yes

4. Have the authors made all data underlying the findings in their manuscript fully available?

Reviewer #3: Yes

5. Is the manuscript presented in an intelligible fashion and written in standard English?

Reviewer #3: Yes

Reviewer #3: The authors have fully addressed all my concerns. The authors have fully addressed all my concerns. The authors have fully addressed all my concerns.

**Do you want your identity to be public for this peer review?** For information about this choice, including consent withdrawal, please see our Privacy Policy

Reviewer #3: No

---

## [Editor Report · Acceptance letter]

PONE-D-25-47575R1

PLOS One

Dear Dr. Techapongsatorn,

I'm pleased to inform you that your manuscript has been deemed suitable for publication in PLOS One. Congratulations! Your manuscript is now being handed over to our production team.

Kind regards,

on behalf of

Dr. Ziyu Qi

Academic Editor

PLOS One